# Effect of Dietary Insoluble and Soluble Fibre on Growth Performance, Digestibility, and Nitrogen, Energy, and Mineral Retention Efficiency in Growing Rabbits

**DOI:** 10.3390/ani10081346

**Published:** 2020-08-04

**Authors:** Carlos Farías-Kovac, Nuria Nicodemus, Rebeca Delgado, César Ocasio-Vega, Tamia Noboa, Ramadan Allam-Sayed Abdelrasoul, Rosa Carabaño, Javier García

**Affiliations:** 1Departamento de Producción Agraria, Escuela Técnica Superior de Ingeniería Agronómica, Agroalimentaria y de Biosistemas, Universidad Politécnica de Madrid, C/Senda del Rey 18, 28040 Madrid, Spain; scarlos_03@hotmail.com (C.F.-K.); nuria.nicodemus@upm.es (N.N.); rebeca_dm85@yahoo.es (R.D.); cesar.ocasio1@gmail.com (C.O.-V.); taminobo31@hotmail.com (T.N.); rosa.carabano@upm.es (R.C.); 2Poultry Department, Faculty of Agriculture, Fayoum University, Fayoum 63514, Egypt; ras05@fayoum.edu.eg

**Keywords:** insoluble fibre, soluble fibre, feed efficiency, whole body and carcass chemical composition, energy nitrogen and mineral balance, fibre digestibility, mucosa morphology, energy nitrogen and mineral retention efficiency, rabbit

## Abstract

**Simple Summary:**

Rabbits, like other herbivores, require a minimal level of insoluble fibre in the diet to warrant an adequate digestive function. Epizootic rabbit enteropathy (ERE) is the main digestive trouble in growing rabbits that increases the use of antibiotics. The increase of soluble fibre (once met insoluble fibre requirements) limits the incidence of ERE and improves nitrogen and energy balance, while high levels of insoluble fibre seem to favour ERE. This study evaluated whether the increase of soluble and insoluble fibre above the current requirements of insoluble fibre had a positive impact on mortality, growth performance, diet digestibility, and energy, nitrogen, and mineral balance. Treatments had no effect on mortality, which was low (1%). The increase of insoluble fibre reduced the dietary digestible energy, while soluble fibre only increased it when combined with a low insoluble fibre level. The group fed with the lowest insoluble and soluble fibre levels showed the best energy and mineral balance, while the increase of insoluble or soluble fibre did not improve any growth trait. We conclude that in healthy rabbits, and once the minimal insoluble fibre requirements are met, no increase of insoluble or soluble fibre is recommended.

**Abstract:**

Dietary soluble fibre limits the incidence of epizootic rabbit enteropathy (ERE) and improves the energy and nitrogen balance in low-insoluble fibre diets, while high-insoluble fibre diets seem to favour ERE. This study assessed whether the positive effects of soluble fibre are influenced by the level of insoluble fibre. Four diets (2 × 2 factorial arrangement) were used with two levels of insoluble fibre (314 vs. 393 g/kg DM) and soluble fibre (87 vs. 128 g/kg DM), resulting in four diets with increasing total dietary fibre levels. Growth performance and chemical composition (body and carcass) (28–62 days of age), faecal digestibility (54–57 days of age), and jejunal morphometry functionality (39 days of age) were determined. Mortality was low (<1%) and treatments did not influence it. Insoluble and soluble fibre tended to reduce the growth rate (*p* ≤ 0.109), body protein, and fat accretion (*p* = 0.049 to 0.120), but only insoluble fibre impaired feed efficiency (*p* < 0.001). The efficiency of digestible energy used for growth was impaired with the increase of total dietary fibre (*p* = 0.027), while that of nitrogen remained majorly unaffected. In conclusion, in healthy rabbits, the increase of either insoluble or soluble fibre had no benefit.

## 1. Introduction

The inclusion of a moderated level of soluble fibre (~12%; SF), usually derived from sugar beet pulp (SBP) inclusion, enhanced gut functionality [1,2] and modified intestinal microbiota [3,4,5], which was related to a decrease of the mortality rate in rabbits affected by epizootic rabbit enteropathy (ERE) [6,7,8]. These positive effects associated with SBP were accounted not only for its SF but also for its high content of easily fermentable insoluble fibre [9].

The inclusion of around 15–25% of SBP is required to meet the recommended SF level. However, the energy and nutrient retention efficiency in growing rabbits seem to be directly affected by the level of fermentable fibre. When SBP was included above 15% in the diet replacing barley, it reduced the energy and nitrogen retention efficiency [10]. Nevertheless, a 30% SBP level of inclusion, in substitution of barley and alfalfa, did not affect these traits in low-fibrous diets [11]. On the opposite, energy and nitrogen retention efficiency increased when a lower level of SBP was used (18%) in low-fibre diets (31% neutral detergent fibre, NDF, on DM basis and free of ash and protein) [8], probably due to the improvement of the rabbit health status and the reduction of urine nitrogen losses. In this context, the increase of the insoluble fibre level (NDF) impaired the nutritive value of SBP [12], which might be related with the role of NDF in modulating the microbial activity and rate of passage [13,14], and finally might affect the dietary energy and nitrogen efficiency. The use of high-insoluble fibrous diets for fattening rabbits is not strange at the field level, even when they might increase the incidence of ERE [15,16].

The aim of this work was to clarify whether the positive effect of the moderated inclusion of SF is influenced or not by the NDF level on growth performance, diet digestibility, gut mucosa morphometry and functionality, and energy, protein and mineral retention efficiency of growing rabbits.

## 2. Materials and Methods

### 2.1. Animals and Housing

A total of 264 crossbred hybrid rabbits (New Zealand White × Californian, V × R line from UPV, Valencia, Spain) weaned at 28 days of age were used. They came from 71 multiparous rabbit does from a farm periodically affected by ERE. Rabbits after weaning received the same diets as those offered to their mothers (described below). Rabbits were individually caged in flat deck cages (610 × 250 × 330 mm), except during the faecal digestibility trial in which they were kept in wire metabolism cages (405 × 510 × 320 mm) that allowed the separation of faeces and urine. Rabbits had *ad libitum* access to feed and water with no antibiotic supplemented throughout the experimental period and their health was checked daily. The temperature ranged between 18 and 23 °C. All the experimental procedures used were in compliance with the Spanish guidelines for care and use of animals in research [17], and authorized by the Dirección General de Agricultura y Ganadería from the Community of Madrid (PROEX 328/15).

### 2.2. Diets

Four diets in a 2 × 2 factorial arrangement were used with two levels of insoluble fibre quantified as NDF (314 vs. 393 g/kg DM, free of ash and protein) and two levels of SF quantified by the difference of total dietary fibre and NDF (87 vs. 128 g/kg DM) (Table 1). A control diet was formulated to meet most nutrient requirements for growing rabbits [18], with an NDF level (344 g NDF free of ash/kg DM) similar to that proposed for post-weaned rabbits [15] (342 g NDF free of ash/kg DM), and with a level of SF lower than the one recommended for growing rabbits under ERE risk [7] (LIF-LSF diet). The increase of NDF was obtained by replacing barley and wheat from the LIF-LSF diet for wheat straw, defatted grape seed meal, dehydrated alfalfa, and lard, the latter to minimize the reduction of digestible energy (DE) content (HIF-LSF diet). The increase of SF was obtained by replacing wheat bran, sunflower meal, gluten feed, wheat straw, and grape seed meal from the LIF-LSF diet for SBP (LIF-HSF diet). A fourth diet high in both NDF and SF was formulated (HIF-HSF diet).

### 2.3. Growth Performance and Body Chemical Composition Trial

In total, 224 rabbits (56/diet) weighing 510 ± 74 g were weaned at 28 days of age, and growth rate, feed intake, feed efficiency, and mortality was recorded until 62 days of age. The whole body and carcass chemical composition and energy content was estimated in vivo using the bioelectrical impedance analysis (BIA) technique in 39 rabbits/diet (randomly selected from this group), weighing 511 ± 77 g. Measurements of resistance and reactance were measured in rabbits with a body composition analyser (Model Quantum II, RJL Systems, Detroit, MI, USA) at 28 and 62 days of age [19,20]. Multiple regression equations were used to estimate water, protein, ash, fat, and energy proportions both in the whole body and in the carcass according to these authors. Twenty-four rabbits were discarded due to an excess of feed waste (5, 2, 9, and 8 rabbits from the LIF-LSF, LIF-HSF, HIF-LSF, and HIF-HSF groups, respectively) and two rabbits died from the LIF-HSF group.

### 2.4. Faecal Digestibility Trial

A group of 40 rabbits (10/diet) weighing 1905 ± 108 g (belonging to the growth performance group) were caged individually in metabolism cages to determine the apparent faecal digestibility of gross energy, protein, and fibre fractions according to Perez et al. [21]. Feed intake and total faecal output were recorded for each rabbit during four consecutive days (from 54 to 57 days of age). Faeces were collected daily and were stored at −20 °C, dried at 80 °C for 48 h, and ground with a 1-mm screen for analysis.

### 2.5. Calculations of Energy and Nitrogen Efficiency

Estimated values for the total whole-body nitrogen and energy content were used to obtain the nitrogen and energy retention in the whole body (NR whole body and ER whole body, respectively) at 28 and 62 days of age. Values were expressed per kg BW^0.75^ and day (where the BW was calculated as the average of the final and the initial body weight). Estimated values for the carcass nitrogen, and energy content were used to calculate the nitrogen and energy retention in the carcass (NR carcass and ER carcass, respectively) at 28 and 62 days of age. Values were also expressed per kg BW^0.75^ and day. Moreover, nitrogen and gross energy intake (Ni and GEi, respectively) and digestible N and DE intake (DNi and DEi, respectively) were recorded to calculate the overall N and GE whole-body retention efficiency as: NR whole body/Ni, NR whole body/DNi, ER whole body/GEi and ER whole body/DEi. Besides, the overall N and GE carcass retention efficiency was calculated as: NR carcass/DNi and ER carcass/DEi.

Total N and GE excretion as skin and viscera, faeces, or heat production and urine were calculated as follows:N lost as skin and viscera (g/kg BW^0.75^ and day) = (g NR in the whole body − g NR in the carcass)/kg BW^0.75^ and day.(1)
N excreted as faeces (g/kg BW^0.75^ and day) = (Ni − DNi)/kg BW^0.75^ and day.(2)
N excreted as urine (g/kg BW^0.75^ and day) = (DNi − NR in the whole body)/kg BW^0.75^ and day.(3)
Energy lost as skin and viscera (MJ/kg BW^0.75^ and day) = (MJ ER in the whole body − MJ ER in the carcass)/kg BW^0.75^ and day.(4)
Energy excreted as faeces (MJ/kg BW^0.75^ and day) = (GEi − DEi)/kg BW^0.75^ and day.(5)
Energy excreted as urine and heat production (MJ/kg BW^0.75^ and day) = (DEi − RE in the whole body)/kg BW^0.75^ and day.(6)

Energy retained as protein and fat was calculated taking into account that energy deposited as protein and fat equals 23.15 and 35.65 kJ/g [22]. Energy retained as fat was also calculated as the difference between the total retained energy and the retained energy as protein. Digestible nitrogen and DE intake used for production was obtained by the difference between the total DNi and DEi and the nitrogen and energy requirements for maintenance, respectively (0.464 g DN/kg BW^0.75^ day, and 430 kJ DE/kg BW^0.75^ day) [22]. Mineral balance was estimated in the same way as the nitrogenous and energetic ones.

### 2.6. Gut Histology and Enzymatic Activity and Immune Function

Another different group of 40 rabbits (10/treatment) of 28 days of age and 472 ± 84 g BW were caged in groups of 2 rabbits (5 cages/treatment) and slaughtered by head concussion at 39 days of age. A sample from the middle part of the jejunum (3 cm) were collected in 10% buffered neutral formaldehyde solution (pH 7.2 to 7.4) for histological analysis and processed as described [23]. Another 6 cm were excised from the middle part of the jejunum, flushed with saline solution, frozen in dry ice, and immediately stored at −80 °C to determine sucrose-specific activity as described [1]. Villous height, crypt depth, and goblet cell from each villous were measured individually. Seven samples were discarded due to poor staining or conservation (3, 1, and 3 from the LIF-HSF, HIF-LSF, and HIF-HSF groups, respectively).

### 2.7. Chemical Analysis

Procedures of the AOAC [24] were used to determine DM (method 934.01), ash (method 942.05), crude protein (method 968.06), ether extract (920.39), starch (amyloglucosidase-α-amylase method; method 996.11), and total dietary fibre (985.29; TDF). Dietary NDF, acid detergent fibre, and acid detergent lignin were determined sequentially using the filter bag system (Ankom Technology, New York, NY, USA) [24,25]. Dietary NDF was determined using a thermo-stable amylase without any sodium sulphite added and corrected for ash and protein. The dietary SF was calculated, as TDF–NDF (both corrected for ash and protein), and no correction for mucin was done in the faecal samples. Gross energy (GE) was measured by an adiabatic bomb calorimeter (model 356, Parr Instrument Company, Moline, IL, USA).

### 2.8. Statistical Analysis

Data of faecal digestibility, growth performance, sucrose activity, and nitrogen and energy balances were analysed as a completely randomized design with the level of NDF, SF, and their interaction as the main sources of variation by using the mixed procedure of SAS (SAS Inst., Cary, NC, USA). Weaning weight was used as a linear covariate for growth traits, whole body and carcass chemical composition, and nitrogen, energy and mineral balances. Mortality was analysed using a logistic model (GENMOD procedure of SAS using a binomial distribution) considering the same variables, and the results were transformed from the logit scale. All data were presented as least-squares means. When interactions were significant, comparisons among all the treatment means were made using a *t*-test.

## 3. Results

In the whole experimental period, rabbits fed with HIF diets showed a higher feed intake (by 11%; *p* < 0.001, Table 2) and tended to decrease the growth rate (*p* = 0.109), leading to a lower feed efficiency (by 12%; *p* < 0.001) compared with the LIF group. In contrast, rabbits fed HSF diets reduced their feed intake (by 2%; *p =* 0.048), with no effect on the feed efficiency. No interaction was observed between NDF and SF in growth traits. Treatments did not affect the mortality rate, which was lower than 1%.

The digestibility of dry and organic matter decreased with the NDF level (by 25% and 14%, respectively; *p* < 0.001. Table 3) but increased with the level of SF (by 3.5% and 4%, respectively; *p* < 0.001). In contrast, the ash digestibility decreased with both the level of NDF and SF (by 10% and 8%, respectively; *p* ≤ 0.014). An interaction between the level of NDF and SF was observed for the digestibility of energy, protein, total dietary fibre, and NDF (*p* ≤ 0.037). Rabbits fed HIF diets had a lower GE and protein digestibility than those fed LIF diets (by 11% and 4%; *p* < 0.001), while the increase of SF improved GE and protein digestibility with LIF but not with HIF diets. The increase of NDF at the low SF level (LIF-LSF vs. HIF-LSF) did not modify the digestibility of any fibre fraction, but the increase of SF improved the digestibility of NDF and SF (*p* < 0.001), being higher in LIF than in HIF groups. These values resulted in very similar ratios of dietary digestible protein/DE among groups, although those of the HSF groups were lower (*p* < 0.001), and that of the HIF-HSF group tended to be the lowest (*p* < 0.070).

At 28 days of age, rabbits from HIF groups showed a higher whole-body fat, and a lower whole-body protein and mineral proportions than those from LIF groups (22.4% vs. 23.4% for fat, 59.1% vs. 58.5% for protein, and 11.7% vs. 11.5% for minerals, all of them on a DM basis; *p* < 0.001. Appendix A), with no differences in the carcass protein concentration (Appendix A). However, an interaction NDF × SF was found for the fat and mineral proportions, being higher for the LIF-HSF and HIF-LSF groups (*p* ≤ 0.062. Appendix A). At 62 days of age, there were no differences in the whole-body fat proportion, but the whole-body protein concentration increased in the HIF and HSF groups (in both by 1%; *p* ≤ 0.032). Additionally, rabbits fed HSF diets increased in the carcass the protein and decreased the fat proportions (by 0.8 and 3%, respectively; *p* ≤ 0.028), and a similar trend was observed for the HIF groups (*p* ≤ 0.067), while the HIF diets tended to increase the carcass mineral proportion (*p* = 0.055). The daily whole-body protein and fat accretion tended to reduce with the increase of NDF and SF (*p =* 0.049 to 0.120, Appendix A), with no effect on mineral accretion. The composition of the body weight gain had lower fat in the HIF than in the LIF groups (*p* = 0.050), and a lower protein content (*p* = 0.034). The daily carcass fat accretion and the fat proportion in the carcass weight gain decreased in the HSF compared to the LSF groups (by 5% and 2%; *p* ≤ 0.039, Appendix A), while the daily carcass protein and mineral accretion and the protein and mineral proportions in the carcass decreased in the HIF compared to the LIF groups (by 4% and 2% for protein and by 3 and 1% form minerals; *p* ≤ 0.060).

The digestible nitrogen intake was higher for the HIF-LSF group compared with the other three groups (2.41 vs. 2.26 g DNi/kg BW^0.75^ d; *p =* 0.003, Table 4). The nitrogen retained in the whole body and in the carcass decreased by 2–3% when NDF increased (*p* ≤ 0.040), with no effect of SF. These results led to the lowest DN retention efficiency in the whole body and in the carcass in the HIF-LSF group compared to the other three groups (0.427 vs. 0.460, and 0.284 vs. 0.310; *p* ≤ 0.011), as well as for the retention efficiency of the estimated DN used for growth (0.506 vs. 0.553, and 0.336 vs. 0.372; *p* ≤ 0.011). Faecal nitrogen losses were lower in the LIF groups, and HSF diets reduced them when combined with the LIF diet but increased with the HIF diet (*p* < 0.001). They were inversely proportional to the digestible crude protein content of the diets. Urinary nitrogen losses were the highest for the HIF-LSF group, showing similar values to the other three groups (1.38 vs. 1.23 g urinary/kg BW^0.75^ d; *p =* 0.002).

The DE intake was also the highest for the HIF-LSF group compared with the other three groups (1.15 vs. 1.10 MJ DE/kg BW^0.75^ d; *p =* 0.027, Table 5). The energy retained in the whole body and in the carcass tended to decrease when NDF and SF increased (*p* ≤ 0.084). Consequently, the LIF-LSF group had the highest DE retention efficiency in the whole body and in the carcass compared with the other three groups (0.330 vs. 0.302, and 0.204 vs. 0.185, *p* ≤ 0.029). These differences were even higher for the retention efficiency of estimated DE used for growth in the whole body and in the carcass, decreasing with the increase of total dietary fibre, although showing the lowest values in rabbits fed the HIF-LSF diet (*p* ≤ 0.027). Faecal energy losses increased with NDF and decreased with SF (*p* < 0.001), and were inversely proportional to the DE of the diets. The energetic losses as urine and heat production were minimal for the LIF-LSF group and maximal for the HIF-LSF group, showing intermediate values for the other two groups (*p =* 0.014).

The digestible mineral intake was the highest for the HIF-LSF group compared with the other three groups (4.59 vs. 4.01 g MIi/kg BW^0.75^ day; *p <* 0.001, Table 6). The minerals retained in the whole body were not affected by treatments, but those retained in the carcass decreased by 2% when NDF increased (1.021 vs. 0.997 g MI/kg BW^0.75^ day; *p <* 0.001). The LIF-LSF group showed the highest efficiency of retention of digestible minerals in the carcass, which decreased with both NDF and SF, although the HIF-LSF group showed the lowest value (*p* < 0.001). Faecal mineral losses increased with both the level of NDF and SF (by 11% and 8%, respectively; *p* < 0.001). Mineral urine losses evolved inversely to the carcass retention efficiency (*p* < 0.001).

The increase of NDF had no influence on the villous height in the LSF groups but tended to decrease it in LIF-HSF and increase it in the HIF-HSF groups (*p =* 0.071, Table 7). Treatments did not affect the crypt depth, the ratio villous height/crypt depth, and the number of goblet cells per villi in the jejunal mucosa. The increase of SF impaired the sucrose specific activity (by 21%; *p* = 0.034), while the increase of NDF tended to reduce it (*p* = 0.059).

## 4. Discussion

Digestive disorders are one of the main causes of mortality in growing rabbits, with antibiotic treatment being the usual way to control them [26]. One of the nutritional strategies to limit ERE is the inclusion of moderate levels of SF [3,6,27,28], while the increase of NDF might increase its incidence [15,16]. However, in this study, a very low mortality rate was reported, avoiding any conclusion about treatments, and suggesting a stronger influence of other environmental factors than the diet. Even the higher dietary protein levels than those expected did not trigger an ERE outbreak, although a high protein level is recognized as an important risk factor [27,29,30,31]. Under these good sanitary conditions, there was no positive effect of the level of SF on the mucosa morphology (ratio villous height/crypth depth and number of goblet cells) or functionality (sucrose-specific activity), which contrast with the positive effect reported when there is an ERE outbreak [1,5]. The trend observed for the interaction in the villous height suggests a negative influence of SF in the LIF groups but positive in the HIF groups, an observation that will require further confirmation. The negative influence of NDF on the sucrose-specific activity, similarly to that of SF, might be partially associated to the reduction of starch intake, as they were positively correlated (*n* = 4; *r* = 0.90; *p* = 0.099), rather than to mucosa damage.

The fibre digestibility depends on microbial activity, fermentation time, and digestion rate, which are all affected by the chemical composition of dietary fibre (mainly by the degree of lignification of NDF and the SF content) and particle size [32]. The digestibility of NDF of the LIF-LSF diet was low and close to other low insoluble-soluble fibre diets with medium-high DE concentrations [3,8,27,28,33]. The increase of NDF when SF was low (HIF-LSF vs. LIF-LSF) did not modify NDF digestibility, although the degree of lignification and feed intake increased, indicating that it probably accelerated the rate of passage [34,35]. This lack of effect might be associated to the fast digestion rate of the degradable fraction of NDF, which can be degraded/solubilized up to 50% before the caecum [9]. Comparatively, it was surprising that soluble fibre digestibility decreased from the LIF-LSF to the HIF-LSF group, which might account for the potential reduction in the mean retention time and the changes in the ingredient proportions. However, a higher incidence of these factors on the NDF than on the SF fraction was expected. The increase of SF when NDF is low (LIF-HSF vs. LIF-LSF) improved both NDF and SF digestibility in agreement with previous studies [7,9,23]. It is explained by the faster digestion rate of SF, similar to that observed in vitro for SBP when compared with straw [36]. The increase of NDF when SF is high (HIF-HSF vs. LIF-HSF) improved also the NDF and SF digestibility as expected [37,38] but the NDF to a lesser extent. It might be linked to the faster rate of passage associated to the HIF groups, which might limit the fermentation of SBP fibre and accordingly its nutritive value. This effect was already observed [12], when a higher DE value for SBP using a high energy basal diet than a low one was reported.

The increase of NDF led to a higher feed intake, and impaired feed efficiency as expected. On the opposite, the increase of SF did not modify feed efficiency, although it tended to reduce feed intake and the growth rate. The replacement of starch mainly by SF and the absence of ERE might be behind this lack of effect on feed efficiency as observed previously [26], although other authors found a positive effect [8,37,39]. These contradictory results are not explained by differences in the acid detergent fibre content [7,40], and do not seem associated to the health status. However, a potential improvement of the nitrogenous retention efficiency, and a reduction of nitrogen losses might be behind the positive effect of SF on feed efficiency [8]. In contrast, the inclusion of SF in the substitution of NDF, and minor proportions of starch and fat, usually improves the feed efficiency [3,38,41]. The magnitude of the feed intake reduction with the inclusion of SF widely differs among studies, with no great influence on DEi in some of them, but in others, where the level of SBP inclusion exceeded 15–30%, compromised DEi (and even DNi), impairing the feed conversion ratio and dressing out percentage [6,8,10,42,43]. It was related to the accumulation of digesta in the caecum derived from a combination of factors like the level and type of NDF, and effective particle size [44].

In this study, rabbits tended to grow slower either with the increase of NDF or SF or both, even when they had a similar or even higher DE and DN intake. These results agree with the trend to reduce the whole-body protein and fat accretion with the increase of total dietary fibre, with the reduction being more pronounced for fat than for protein. It led to a progressive reduction of the nitrogen and energy retained in the whole body and in the carcass, even when diets were formulated to meet all nutrient requirements, which contrasts with the lack of an effect of SF on these traits observed in a previous study [8]. These results are explained by the trend to impair the retention efficiency of the DE used for growth (DEp) both in the whole body and in the carcass when total dietary fibre increased. The reduction of the energetic efficiency is parallel to the increase of energetic losses as urine and heat production from the LIF-LSF to the HIF-HSF group (excluding the HIF-LSF group), most probably related to the heat increment associated to the use of nutrients for growth and that associated with the increase of the fermentative activity, as the N losses in urine were similar among these groups. The increase of total dietary fibre (from the LIF-LSF to the HIF-HSF group) implied a higher intake of digestible total dietary fibre (by 65%) and a lower intake of starch (by 55%), accompanied in the HIF groups by an increase of fat intake (by 140%). The higher metabolic efficiency of glucose compared to volatile fatty acids, and the potential need in the HIF groups to obtain glucose from amino acids seemed to be behind the energetic efficiency impairment observed with the increase of total dietary fibre. In relation to fat, other authors found no influence of the increase of fat intake on the overall DE or DE efficiency for growth [45], suggesting a minor influence on the energetic efficiency results of this study. The impairment of the efficiency of the retention of DE (used for growth) was not related with the efficiency of retention of DN (used for growth), which was similar among groups (excluding the HIF-LSF group, which will deserve a separate comment). These results contrast with the lack of an effect of SF on the retained energy and nitrogen observed previously [8], which might be explained by the higher starch and lower fermentable fibre intake obtained in their study [8,23]. Besides, the same authors observed a positive effect of the increase of SF on nitrogen metabolism due to the reduction of urinary nitrogen losses, probably derived from a shift of urinary nitrogen excretion to faecal excretion. This effect was not observed in this study probably due to the excess of dietary digestible protein, despite the increase of the fermentable fibre intake with the level of total dietary fibre. Rabbits fed the HIF-LSF diet showed the worst efficiencies of DN and of DE used for growth, which were lower than those expected according to its NDF and SF levels. It is mainly explained by the high DN intake, derived from the combination of the highest feed intake and a high dietary DN/DE ratio, that led to the highest urinary nitrogen (and energetic) losses. The latter are associated with higher heat production derived from urea synthesis, which helps to explain the trend to increase energetic losses as urine and heat production in rabbits fed HIF-LSF compared with those of the HIF-HSF group (*p* = 0.112). In fact, when experiments with wide differences in dietary digestible protein content are combined, a close relationship between the DN intake and urinary nitrogen losses is found (Figure 1).

The influence of dietary fibre on mineral balance in rabbits has been scarcely studied. The mineral retention in the carcass was impaired with the increase of NDF, although the digestible mineral intake was similar (HIF-HSF) or even higher (HIF-LSF group). It led to a reduction on the efficiency of retention of digested minerals with the increase of total dietary fibre (except for the HIF-LSF group). This result agreed with the reduction of the serum mineral concentrations (per unit of mineral ingested) in pregnant sows when dietary fibre increased [46]. However, the increase of dietary fibre did not influence all minerals in the same way. In fact, the apparent absorption of calcium, phosphorous, and magnesium was not influenced by the fibre level, while it impaired that of sodium and potassium in pigs [47]. In rabbits, the type of fibre (alfalfa, olive pulp, grape pulp) also influenced the apparent absorption of different minerals, but no effect was reported on the plasma levels of most elements [48].

## 5. Conclusions

The increase of either NDF or SF in diets for healthy rabbits had no positive effect in terms of growth traits, retention efficiency of digestible nitrogen energy and minerals, and mucosa morphology, once the NDF requirements for growing rabbits are met. Further research is required to identify what is the optimal level of NDF to combine with SF under an enteropathy outbreak.

## Figures and Tables

**Figure 1 animals-10-01346-f001:**
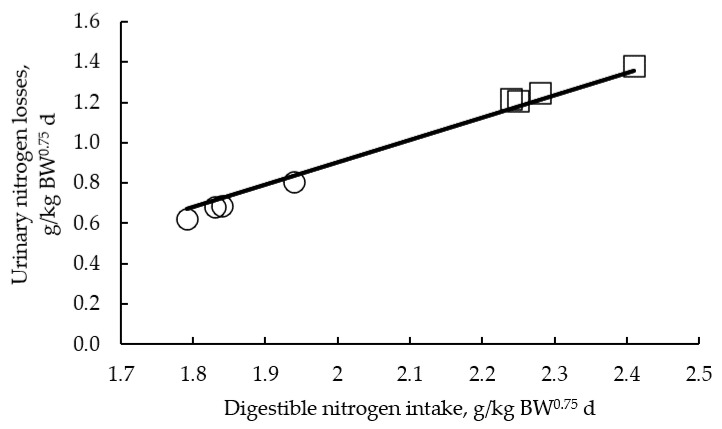
Relationship between digestible nitrogen intake and urinary nitrogen losses during fattening in rabbits (Urinary nitrogen losses = −1.31(±0.13) + 1.11(±0.061) digestible nitrogen intake; N = 8; *p* < 0.001. Original observations plotted with the mean regression line across studies: ○ Delgado et al., 2018. □ Current study).

**Table 1 animals-10-01346-t001:** Ingredients and chemical composition of the experimental diets.

Item ^1^	Low-Insoluble Fibre		High-Insoluble Fibre	
	Low SF	High SF	Low SF	High SF
	LIF-LSF	LIF-HSF	HIF-LSF	HIF-HSF
Ingredient, g/kg as fed				
Barley	150.0	141.0	70.0	70.0
Wheat	150.0	141.0	70.0	70.0
Dehydrated alfalfa	101.8	101.8	154.0	154.0
Cereal straw	84.0	63.0	169.3	146.6
Defatted grape seed meal	36.0	27.0	72.6	62.9
Wheat bran	176.5	60.0	133.8	10.0
Sugar beet pulp	0.0	170.0	0.0	170.0
Gluten feed	93.5	30.5	71.3	5.0
Sunflower meal	110.0	171.0	130.0	185.0
Soybean meal	60.0	60.0	60.0	60.0
Lard	0.0	0.0	35.0	35.0
Cane molasses	10.0	10.0	10.0	10.0
L-Lysine 50	3.0	2.4	3.2	2.6
Alimet	0.5	0.5	0.9	0.9
L-Threonine	1.0	0.6	1.2	0.8
L-Tryptophan	0.2	0.2	0.2	0.2
Sodium chloride	2.5	2.5	2.5	2.5
Calcium carbonate	16.0	8.0	10.0	4.0
Calcium phosphate	3.5	9.0	4.5	9.0
Mineral/vitamin premix ^1^	1.5	1.5	1.5	1.5
Chemical composition, g/kg DM				
DM	903	905	908	906
Ash	83.9	89.1	90.9	96.3
Total dietary fibre ^2^	398	444	484	519
Neutral detergent fibre (NDF ^2^)	311	317	396	390
Acid detergent fibre ^3^	162	179	230	247
Acid detergent lignin ^3^	36.6	38.3	66.4	65.2
Soluble fibre ^4^	86.7	127.5	88.1	129.0
Crude protein (CP)	197	198	187	183
CP-Total dietary fibre	61.1	67.9	69.2	72.3
CP-NDF	33.8	37.4	33.8	38.0
Starch	212	181	117	86.3
Ether extract	31.6	35.9	79.3	65.7
Gross energy, MJ/kg DM	18.0	17.7	18.9	18.3

^1^ Mineral and vitamin composition (per kg of complete diet): 20.0 mg of Mn; 60 mg of Zn; 12 mg of Cu; 1.01 mg of I; 0.30 mg of Co; 42 mg of Fe; 0.12 mg of Se; 9750 UI of vitamin A; 1200 UI of vitamin D_3_, 42 UI of vitamin E, 1.5 mg of vitamin K_3_; 1.5 mg of vitamin B_1_; 3.75 mg of vitamin B_2_; 1.5 mg of vitamin B_6_; 0.018 mg of vitamin B_12_; 12 g of Pantothenic acid; 40 g of Nicotinic acid; 0.75 mg of Folic acid; 0.075 mg of Biotin. ^2^ Corrected for ash and protein. ^3^ Corrected for ash. ^4^ Quantified as total dietary fibre NDF (both corrected for ash and protein).

**Table 2 animals-10-01346-t002:** Effect of dietary level of insoluble and soluble fibre on rabbits’ growth performance from 28 to 62 days of age.

Item	Diets ^1^				SEM		*p*-Value		
	Low-Insoluble Fibre		High-Insoluble Fibre		IF and SF	IF × SF	IF	SF	IF × SF
	Low-Soluble Fibre LIF-LSF	High-Soluble Fibre LIF-HSF	Low-Soluble Fibre HIF-LSF	High-Soluble Fibre HIF-HSF					
N	51	52	47	48					
28–42 days of age									
Body weight 28 day, g	496	459	533	540	6.58	9.30	<0.001	0.103	0.018
Growth rate, g/day	52.5	51.1	52.6	52.0	0.66	0.93	0.660	0.266	0.653
Feed intake, g/day	107	107	118	118	1.89	2.68	<0.001	0.907	0.974
Feed efficiency, g/g	0.519	0.489	0.447	0.445	0.010	0.015	<0.001	0.277	0.340
42–62 days of age									
Body weight 42 day,g	1241	1221	1242	1233	9.172	13.01	0.656	0.263	0.649
Growth rate, g/day	50.7	50.3	49.2	47.9	0.50	0.71	0.012	0.234	0.517
Feed intake, g/day	152	148	170	164	1.51	2.15	<0.001	0.009	0.515
Feed efficiency, g/g	0.336	0.345	0.291	0.294	0.004	0.005	<0.001	0.276	0.581
28–62 days of age									
Body weight 62 g/day	2254	2227	2225	2191	13.39	19.01	0.109	0.098	0.867
Growth rate, g/day	51.4	50.6	50.6	49.6	0.39	0.56	0.109	0.099	0.870
Feed intake, g/day	133	131	149	145	1.22	1.73	<0.001	0.048	0.650
Feed efficiency, g/g	0.388	0.391	0.341	0.344	0.003	0.005	<0.001	0.541	0.968

^1^ LIF-LSF = Low-insoluble fibre, Low-soluble fibre; LIF-HSF = Low-insoluble fibre, High-soluble fibre HIF-LSF = High-insoluble fibre, Low-soluble fibre; HIF-HSF = High-insoluble fibre, High-soluble fibre. Weight at weaning (28 days) as a covariate.

**Table 3 animals-10-01346-t003:** Effect of dietary level of insoluble and soluble fibre on faecal apparent digestibility of dietary components in rabbits from 54 to 57 days of age.

Item	Diets ^1^				SEM		*p*-Value		
	Low-Insoluble Fibre		High-Insoluble Fibre		IF and SF	IF × SF	IF	SF	IF × SF
	Low-Soluble Fibre LIF-LSF	High-Soluble Fibre LIF-HSF	Low-Soluble Fibre HIF-LSF	High-Soluble Fibre HIF-HSF					
N	9	8	10	9					
Initial body weight, g	1961	1856	1939	1904	23.26	32.88	0.691	0.042	0.300
Feed intake, g DM/day	135	128	136	142	3.28	4.63	0.113	0.803	0.180
Faecal apparent digestibility, %									
Dry matter	63.9	66.8	55.7	57.0	0.40	0.57	<0.001	<0.001	0.111
Organic matter	65.1	68.5	56.6	58.1	0.44	0.62	<0.001	<0.001	0.147
Ash	50.5	48.3	47.5	41.7	1.08	1.53	0.004	0.014	0.237
Gross energy	63.1 ^b^	66.7 ^a^	56.2 ^c^	57.2 ^c^	0.41	0.58	<0.001	<0.001	0.037
Crude protein, CP	75.7 ^ab^	76.8 ^a^	74.5 ^b^	72.1 ^c^	0.49	0.70	<0.001	0.377	0.018
Total dietary fibre	35.4 ^c^	50.1 ^a^	33.5 ^c^	40.9 ^b^	0.59	0.83	<0.001	<0.001	<0.001
Neutral detergent fibre	28.1 ^c^	40.2 ^a^	28.0 ^c^	31.6 ^b^	0.69	0.97	<0.001	<0.001	<0.001
Soluble fibre	61.9	74.7	58.0	69.0	1.51	2.14	0.034	<0.001	0.637
Digestible energy, MJ/kg DM	11.38 ^b^	11.78 ^a^	10.64 ^c^	10.46 ^c^	0.075	0.106	<0.001	0.287	0.010
Digestible CP, g/kg DM	14.91 ^a^	15.23 ^a^	13.89 ^b^	13.17 ^c^	0.095	0.134	<0.001	0.147	<0.001
Ratio digestible [CP/energy], g/MJ	13.11	12.93	13.105	12.59	0.052	0.074	0.012	<0.001	0.069

^1^ LIF-LSF = Low-insoluble fibre, Low-soluble fibre; LIF-HSF = Low-insoluble fibre, High-soluble fibre HIF-LSF = High-insoluble fibre, Low-soluble fibre; HIF-HSF = High-insoluble fibre, High-soluble fibre. ^a–c^ Fattening mean values in the same row with a different superscript differ *p* < 0.050.

**Table 4 animals-10-01346-t004:** Effect of dietary level of insoluble and soluble fibre on nitrogen (N) balance from 28 to 62 days of age.

Item	Diets ^1^				SEM		*p*-Value		
	Low-Insoluble Fibre		High-Insoluble Fibre		IF and SF	IF × SF	IF	SF	IF × SF
	Low-Soluble Fibre LIF-LSF	High-Soluble Fibre LIF-HSF	Low-Soluble Fibre HIF-LSF	High-Soluble Fibre HIF-HSF					
N	34	36	32	33					
kg BW^0.75^	1.27	1.25	1.25	1.24	0.01	0.01	0.101	0.086	0.389
^2^ Nitrogen intake, g/kg BW^0.75^ d									
Ni,	2.98	2.97	3.23	3.10	0.03	0.04	<0.001	0.111	0.157
DNi	2.25 ^b^	2.28 ^b^	2.41 ^a^	2.24 ^b^	0.02	0.03	0.138	0.030	0.003
^3^ Nitrogen retained, g/kg BW^0.75^ d									
NR whole body	1.05	1.03	1.02	1.02	0.05	0.01	0.040	0.166	0.510
NR carcass	0.711	0.696	0.682	0.677	0.007	0.010	0.017	0.275	0.570
^4^ Nitrogen efficiency									
NR whole body/Ni	0.353	0.351	0.318	0.330	0.004	0.006	<0.001	0.383	0.186
NR whole body/DNi	0.467 ^a^	0.456 ^a^	0.427 ^b^	0.458 ^a^	0.005	0.007	0.017	0.161	0.005
NR whole body/DNp	0.561 ^a^	0.549 ^a^	0.506 ^b^	0.550 ^a^	0.007	0.010	0.015	0.111	0.049
NR carcass/DNi	0.316 ^a^	0.308 ^a^	0.284 ^b^	0.305 ^a^	0.004	0.006	0.004	0.283	0.011
NR carcass/DNp	0.381 ^a^	0.370 ^a^	0.336 ^b^	0.366 ^a^	0.005	0.008	0.004	0.205	0.008
^5^ Nitrogen losses, g/kg BW^0.75^ d									
Skin and viscera	0.337	0.335	0.343	0.342	0.006	0.008	0.469	0.803	0.988
Faeces	0.725 ^c^	0.691 ^d^	0.824 ^b^	0.865 ^a^	0.008	0.011	<0.001	0.74	<0.001
Urine	1.21 ^b^	1.25 ^b^	1.38 ^a^	1.22 ^b^	0.02	0.03	0.057	0.082	0.002

^1^ LIF-LSF = Low-insoluble fibre, Low-soluble fibre; LIF-HSF = Low-insoluble fibre, High-soluble fibre HIF-LSF = High-insoluble fibre, Low-soluble fibre; HIF-HSF = High-insoluble fibre, High-soluble fibre. ^2^ Ni (g Ni/kg BW^0.75^ day): Nitrogen intake. DNi (g DNi/kg BW^0.75^ day): Digestible N intake. ^3^ NR whole body (g/kg BW^0.75^ day): retained N in the whole body. NR carcass (g/kg BW^0.75^ day): retained N in carcass. ^4^ DNp (g DNp/kg BW^0.75^ day): Intake of digestible N used for growth, obtained by the difference between DNi and the DN used for maintenance (0.464 g DNm/kg BW^0.75^ day). Skin and viscera (g N/kg BW^0.75^ day): (g N retained in the whole body − g retained in carcass)/kg BW^0.75^ day. Faeces (g/kg BW^0.75^ day): (Total N intake − DNi)/kg BW^0.75^ day. ^5^ N lost as skin and viscera (g/kg BW^0.75^ day) = (g NR in the whole body − g NR carcass)/kg BW^0.75^ day. N excreted as faeces (g/kg BW^0.75^ day) = (Ni − DNi)/kg BW^0.75^ day. Urine (g/kg BW^0.75^ day): (DNi − NR whole body)/kg BW^0.75^ day. Weight at weaning (28 days) as a covariate. ^a–d^ Fattening mean values in the same row with a different superscript differ *p* < 0.050.

**Table 5 animals-10-01346-t005:** Effect of dietary level of insoluble and soluble fibre on energy (E) balance from 28 to 62 days of age.

Item	Diets ^1^				SEM		*p*-Value		
	Low-Insoluble Fibre		High-Insoluble Fibre		IF and SF	IF × SF	IF	SF	IF × SF
	Low-Soluble Fibre LIF-LSF	High-Soluble Fibre LIF-HSF	Low-Soluble Fibre HIF-LSF	High-Soluble Fibre HIF-HSF					
N	34	36	32	33					
BW^0.75^ d	1.27	1.25	1.25	1.24	0.01	0.01	0.101	0.086	0.389
^2^ Energy intake, MJ/kg BW^0.75^ d									
GEi	1.70	1.65	2.05	1.94	0.02	0.03	<0.001	0.002	0.24
DEi	1.08 ^b^	1.10 ^b^	1.15 ^a^	1.11 ^ab^	0.01	0.02	0.016	0.619	0.027
^3^ Energy retained, kJ/kg BW^0.75^ d									
ER whole body	354	339	338	333	4.04	5.73	0.068	0.067	0.405
ER carcass	219	209	206	203	2.83	4.01	0.030	0.084	0.348
^4^ Energy efficiency ^1^									
ER whole body/GEi	0.208	0.207	0.165	0.173	0.003	0.004	<0.001	0.415	0.247
ER whole body/DEi	0.330 ^a^	0.311 ^b^	0.294 ^b^	0.302 ^b^	0.004	0.006	<0.001	0.350	0.029
ER whole body/DEp	0.677 ^a^	0.631 ^ab^	0.557 ^c^	0.598 ^bc^	0.014	0.020	<0.001	0.896	0.027
ER carcass/DEi	0.204 ^a^	0.191 ^b^	0.179 ^b^	0.184 ^b^	0.003	0.004	<0.001	0.323	0.024
ER carcass/DEp	0.419 ^a^	0.387 ^a^	0.339 ^b^	0.365 ^ab^	0.009	0.012	<0.001	0.831	0.017
^5^ Energy losses, MJ/kg BW^0.75^ d									
Skin and viscera,	0.135	0.132	0.131	0.130	0.003	0.004	0.647	0.331	0.794
Faeces	0.628	0.551	0.896	0.828	0.007	0.010	<0.001	<0.001	0.624
Urine + heat production	0.721 ^c^	0.765 ^bc^	0.814 ^a^	0.777 ^ab^	0.012	0.017	0.004	0.837	0.014

^1^ LIF-LSF = Low-insoluble fibre, Low-soluble fibre; LIF-HSF = Low-insoluble fibre, High-soluble fibre HIF-LSF = High-insoluble fibre, Low-soluble fibre; HIF-HSF = High-insoluble fibre, High-soluble fibre. ^2^ GEi: Gross Energy intake (MJ/kg BW^0.75^ day). DEi: Digestible Energy intake (MJ/kg BW^0.75^ day). ^3^ ER whole body (kJ/kg BW^0.75^ day): GE retained in the whole body. ER carcass (kJ/kg BW^0.75^ day): GE retained in carcass. ^4^ DEp (MJ/kg BW^0.75^ day): Intake of DE used for growth, obtained by the difference between DEi and the DE used for maintenance (430 kJ DE/kg BW^0.75^ day). ^5^ Skin and viscera (MJ/kg BW^0.75^ day): (MJ GE retained in the whole body − MJ GE retained in carcass)/kg BW^0.75^ day. Faeces (MJ/kg BW^0.75^ and day): (GEi − DEi)/kg BW^0.75^ day. Urine + heat production (MJ/kg BW^0.75^ d): (DEi − GE retained in carcass − GE lost in skin and viscera)/kg BW^0.75^ day. Weight at weaning (28 days) as a covariate. ^a–c^ Fattening mean values in the same column with a different superscript differ *p* < 0.050.

**Table 6 animals-10-01346-t006:** Effect of dietary level of insoluble and soluble fibre on mineral (MI) balance from 28 to 62 days of age.

Item	Diets ^1^				SEM		*p*-Value		
	Low-Insoluble Fibre		High-Insoluble Fibre		IF and SF	IF × SF	IF	SF	IF × SF
	Low-Soluble Fibre LIF-LSF	High-Soluble Fibre LIF-HSF	Low-Soluble Fibre HIF-LSF	High-Soluble Fibre HIF-HSF					
N	34	36	32	33					
^2^ Mineral intake, g/kg BW^0.75^ d									
MIi,	7.92	8.34	9.66	9.66	0.09	0.13	<0.001	0.097	0.095
DMIi	3.97 ^b^	4.04 ^b^	4.59 ^a^	4.02 ^b^	0.04	0.06	<0.001	<0.001	<0.001
^3^ Mineral retained, g/kg BW^0.75^ d									
MIR whole body	1.25	1.22	1.23	1.23	0.01	0.01	0.337	0.223	0.068
MIR carcass	1.027	1.016	0.998	0.996	0.005	0.007	<0.001	0.315	0.531
Minerals efficiency									
MIR whole body/MIi	0.159 ^a^	0.148 ^b^	0.127 ^c^	0.128 ^c^	0.002	0.002	<0.001	0.032	0.009
MIR whole body/DMIi	0.318 ^a^	0.306 ^a^	0.268 ^b^	0.308 ^a^	0.003	0.005	<0.001	0.005	<0.001
MIR in carcass/DMIi	0.261 ^a^	0.254 ^ab^	0.218 ^c^	0.249 ^b^	0.003	0.004	<0.001	0.036	<0.001
^4^ Mineral losses, g/kg BW^0.75^ d									
Skin and viscera,	0.227	0.207	0.227	0.236	0.007	0.011	0.192	0.594	0.160
Faeces	3.95	4.30	5.07	5.63	0.05	0.07	<0.001	<0.001	0.110
Urine	2.71 ^b^	2.81 ^b^	3.36 ^a^	2.79 ^b^	0.04	0.06	<0.001	<0.001	<0.001

^1^ LIF-LSF = Low-insoluble fibre, Low-soluble fibre; LIF-HSF = Low-insoluble fibre, High-soluble fibre HIF-LSF = High-insoluble fibre, Low-soluble fibre; HIF-HSF = High-insoluble fibre, High-soluble fibre. ^2^ MIi (g MIi/kg BW^0.75^ day): Mineral intake. DMIi (g DMIi/kg BW^0.75^ day): Digestible mineral intake. ^3^ MIR whole body (g/kg BW^0.75^ day): Mineral retained in the whole body. MIR carcass (g/kg BW^0.75^ day): Mineral retained in carcass. ^4^ Skin and viscera (g Min/kg BW^0.75^ day): (g MIR in the whole body − g MIR in carcass)/kg BW^0.75^ day. Faeces (g/kg BW^0.75^ day): (MIi − DMIi)/kg BW^0.75^ d. Faeces (g/kg BW^0.75^ day) = (MINi − DMIi)/kg BW^0.75^ day. Urine (g/kg BW^0.75^ day): (DMIi − MIR whole body)/kg BW^0.75^ day. Weight at weaning (28 day) as a covariate. ^a–c^ Fattening mean values in the same row with a different superscript differ *p* < 0.050.

**Table 7 animals-10-01346-t007:** Effect of dietary level of insoluble and soluble fibre on mucosa histology and enzyme activity of 39-day-old rabbits.

Item	Diets ^1^				SEM		*p*-Value		
	Low-Insoluble Fibre		High Insoluble Fibre		IF and SF	IF × SF	IF	SF	IF × SF
	Low-Soluble Fibre LIF-LSF	High-Soluble Fibre LIF-HSF	Low-Soluble Fibre HIF-LSF	High-Soluble Fibre HIF-HSF					
N	10	7	9	7					
Villous height, µm	380	340	383	422	14.8	20.8	0.052	0.995	0.071
Crypt depth, µm	144	156	147	160	5.45	7.69	0.667	0.115	0.989
Ratio villous height/crypt depth	2.71	2.20	2.67	2.70	0.15	0.22	0.293	0.290	0.232
Goblet cells, n^o^/villous	11.1	9.86	10.3	11.8	1.06	1.49	0.707	0.923	0.350
Mucose protein, mg/g of tissue	61.5	46.2	57.6	52.2	4.81	6.80	0.878	0.136	0.470
Sucrose activity, µmol of glucose/mg of protein	273	212	218	174	17.0	24.0	0.059	0.034	0.703

^1^ LIF-LSF = Low-insoluble fibre, Low-soluble fibre; LIF-HSF = Low-insoluble fibre, High-soluble fibre HIF-LSF = High-insoluble fibre, Low-soluble fibre; HIF-HSF = High-insoluble fibre, High-soluble fibre.

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
