# Peer review of "Effect of Dietary Insoluble and Soluble Fibre on Growth Performance, Digestibility, and Nitrogen, Energy, and Mineral Retention Efficiency in Growing Rabbits"

_animals, 2020, doi:10.3390/ani10081346_

Round 1
Reviewer 1 Report
The authors revised and corrected the original draft of the manuscript, however, further corrections are required.
L34-35: p-values are reported with two digits in the text, whereas in the tables p-values are reported with three digits. Please check the p-values in all the parts of the manuscript and make uniform.
Table 1: please check the ingredient list of HIF-HSF diets. The sum of ingredients exceeds 1000.
L177: "higher feed intake", compared to?
L178: please check the p-value.
L234: higher... copared to the..
L237: higher... copared to the..
L245: higher... copared to the..
L292-293: a "." il lacking at the end of the sentence.
L305: "r" or r2 ?
L305: p =
L372: "higher energetic losses", compared to?
Table 3: The error term should be reported with the same number of decimals of the mean.
Table 6: The error term should be reported with the same number of decimals of the mean (see the row "MIR carcass", column "IF x SF").
Table 7: The error term should be reported with the same number of decimals of the mean.
Tables S1, S2, S3, S4: The error term should be reported with the same number of decimals of the mean.
In my opinion the Figure 1 is not necessary. In any case, it should be revised. In particular, the font size must be reduced.
Author Response
Dear Madam/Sir,
thank you for your corrections. We have some doubts about the figures/decimals of SEM and we did our best. Enclosed please find a file with comments to the corrections.
Sincerely yours,
Javier García

Reviewer 2 Report
The authors have addressed concerns appropiately.
Author Response
Thank you.
This manuscript is a resubmission of an earlier submission. The following is a list of the peer review reports and author responses from that submission.
Round 1
Reviewer 1 Report
The aim of the study was to assess the impact of different levels of fiber on the growth and health status of young rabbits. The main advantage of study was to demonstrate that if the health status of the total animal population being assessed is high, the impact of fiber levels in feed is of secondary importance for the factors assessed.
Author Response
We agree with the reviewer. Thank you for your comments.
Reviewer 2 Report
In my opinion before the manuscript could be reviewed the authors must re-evaluate the whole experimental dataset.
First of all, the experimental design is not a 2x2 factorial arrangement (line 75). In fact the used experimental pellets differed not only the soluble and insoluble fibre content. Based on the chemical composition given in table 1 we can see that the groups differed also in the starch and either extract content which are significantly influence the examined traits. The authors used the mixed procedure with the insoluble and soluble fibre content as fixed factor. Instead of this method I suggest the usage of the repeated measures of SAS because based on the table 2 we can see that the measurements were performed several times (but this information is missing in the materials section), I also suggest that the diet (1-4) should be the fixed effect (the reason see above). Based on this I suggest that the authors re-evaluate the whole dataset before resubmission.
Moreover, L77-81: “A control diet was formulated to meet most nutrient requirements for growing rabbits [18], but with a lower insoluble fibre content (NDF), similar to that proposed previously [15] (344 vs. 342 g NDF free of ash/kg DM) and with a level of soluble fibre lower than that recommended for growing rabbits under ERE risk [7] (LIF-LSF diet).” – The meaning of this sentence is that you have no control diet which would be indispensable.
Author Response
We understand and appreciate the comments made by the reviewer, which in fact raises interesting questions like that of control group. We have justified in the attached file our position. The changes proposed would change some pre-planned analysis, that are usually accepted.

Reviewer 3 Report
Farías-Kovac and co-authors report on the effect of dietary insoluble and soluble fibre on growth performance, digestibility and nitrogen, energy and mineral retention efficiency in growing rabbits. The study is performed in rabbits born in a farm that has periodic episodes of epizootic rabbit enteropathy (ERE) finding that in healthy rabbits free of ERE, diets with minimal insoluble fibre and low soluble fibre maximize growth rate and nitrogen and energy efficiencies.
The manuscript is clear and well written.
MINOR COMMENTS
Line 59. Replace “rabbits is no strange” with “rabbits is not strange”
Line 68. Replace “the same diets than that offered” with “the same diets than those offered”
Line 71. The Celcius degree symbol is incorrect.
Table 1. I beleive that in the heading it should be Low SF HIF-LSF and High SF HIF-HSF instead of SF HI-LSF and High SF HI-HSF.
Line 118. It is the first time GE is used and not explained. However in line 158 it is explained Gross energy (GE). Please change abbreviation with explanation to the first position.
Line 144. Replace “Jejunum were collected” with “jejunum was collected”
Line 285. Replace “there were no positive effect” with “there was no positive effect”.
MAJOR COMMENTS
The farm the rabbits came from was periodically affected by epizootic rabbit enteropathy and 264 rabbits weaned at the age of 28 d were used and maintained upto 62 d of age. No sign of ERE was observed throughout the study in any of the animals?
Materials and Methods
I’m not sure I understand how many animals were used in total. In line 66 authors state the 264 rabbits were used. In line 95 it says 224, 56 per diet. In line 103 it authors state that 24 rabbits were discarded due to en excess of feed wast but authors do not mention to which diet they belong to (this must be clarified) and then the two rabbits from diet LIF-HSF died. Then in line 106, 40 rabbits were used for faecal digestibility and in line 142 another 40 for gut histology. This is 224+40+40=304. Even if we substract the discarded and deaths it does not make up for the total 264 stated in line 66. Please clarify.
Conclusions
Line 382-383. Please rephrase. I do not understand. Are authors proposing that further research should be done in a farm with no history of enteropathy? If so the sentence should be re-written.
References
Some references are incomplete or not in the correct format.
- check paper or book title
- check journal name
38 and 39. check page numbers
Author Response
Thank you for your corrections. In the file attached are the answers and the corrections made.

Reviewer 4 Report
The Authors aimed to assess the influence of dietary insoluble and soluble fibre on growth performance, digestibility and nitrogen, energy and mineral retention efficiency in growing rabbits. The manuscript was properly conducted and findings reported are important for rabbit production and nutrition. The paper contains important data growth performances of rabbits under different dietary treatments. The Authors investigated an interesting topic and the objective of the paper is of worldwide interest and fits well within the scope of the journal. Results were properly reported and the results have been accurately discussed and compared with other recently published papers.
So, based on my opinion the manuscript merits the acceptance in Animals.
Author Response
Thank you for your comments.
Reviewer 5 Report
The effect of dietary insoluble and soluble fibre on gut health, growth performance and diet digestibility in growing rabbits is widely investigated, however, some new information are given by the present study.
The work is well organized and developed, the experimental design is appropriate as well as the statistical analysis of the data. The results are interesting, but their presentation and discussion should be revised. Moreover, in my opinion, the text should be revised by an English language native speaker.
Specific comments
L6-7: please check the superscript numbers
L12-23: the simple summary should be revised. The introduction could be shortened and some information on diet digestibility could be added
L12: “Rabbits, like other herbivores, require”
L16: “seem”
L18: the term “health status” should be avoided and substituted by “mortality”
L18: please add “diet digestibility”
L18: “growth performance”
L20: “the lowest insoluble and…”
L22: please delete “free of ERE”
L24-36: the summary should be revised. The main results, including those on diet digestibility, should be presented numerically, not only in terms of p-values
L25: “improves”
L26: “seem”
L29: “Growth performance”
L31: “morphometry and functionality”
L32: “p = 0.11”. When a precise level of the probability is reported the authors should use “p =” not “p <” or “p ≤”. Please revise the p-values in all the manuscript. Anyway, p = 0.11 is over the level which is generally accepted as a difference approaching significance. Moreover, I suggest to the authors of avoiding the presentation and discussion of the data that tend to be statistically different, either in the abstract or in the sections of results and discussion
L43-63: please add some information concerning the effect of dietary soluble and soluble fibre on growth performance and gut physiology of growing rabbits. I think that other relevant references (including reviews) could be used, especially for the topic of dietary fibre and gut health
L47: “, but also for its high content of easily fermentable insoluble fibre”
L53: “efficiency increased”
L55: probably due to”
L57 “in modulating”
L62: “on growth performance, diet digestibility, gut mucosa morphometry and functionality, and protein and mineral retention efficiency of growing rabbits"
L64: Please clarify how often you checked health of the rabbits. I think that the main slaughter results (such as chilled carcass weight, reference carcass weight, dressing out percentage and full gut incidence) would be of interest in this study.
L68: please delete “than that”
L69: please report cage dimensions
L76: “soluble fibre (SF)”. The abbreviation SF could be used in all the manuscript.
L78-81: “lower NDF and SF contents than those recommended for growing rabbits under ERE risk”
L85: “from LIF-LSF”
L103: “Twenty four rabbits were discarded”, how many per group?
L107: please report cage dimensions
L110: please add the reference of the harmonised procedures for in vivo digestibility trials in growing rabbits
L143: please report cage dimensions
L148: “Seven samples were discarded”, how many per group?
L172: “p <”
L172: “Table 2), and lower feed efficiency”
L174: please delete “tended to reduce... (p = 0.099)”.
L176-177: please delete “As a result…” this is a discussion of the result
L189: please delete “although that of … (p < 0.070).”
L191: “on rabbits growth performance from 28 to 62 d of age”
L196: “p <”
L203: “p =”
L205 “p =”
L2010-2011: please delete “but not relevant”
L253: “p <”
L260: “p <”
L266: “p <”
L276-373: In my opinion, the discussion should be reorganised. In details, the authors should discuss the results following the order of their presentation, i.e. growth performance, diet digestibility, nitrogen and energy balance, mucosa histology and enzymatic activity
L278: delete “(Solanas et al., 2019)”
L280: delete “(Gutiérrez et al., 2002; Tazzoli et al., 2015)”
L298 “fibre was low”
L301 “before the caecum”
L301-305: please check this sentence. If the mean retention time decreased diet digestibility increased.
L308-309: please delete “, as the feed intake… affected.”
L312-313: please check this sentence. It is not clear for me
L317-318: Please check this sentence. You should clarify that this comment regards the results of other studies
L324: “several factors are implied” such as?
L324: “The reduction of feed intake”
L326: delete “also”
L326-328: This sentence could be deleted
L339-340: “implied a higher intake of digestible total dietary fibre (by 65%) and a lower intake of starch (by 55%), accompanied in...”
L350: is the reference 23 necessary?
L351: “Besides, the same authors, …”
L354: “, despite the increase”
L356: “growth, which were lower”
L365: “reduction on the efficiency”
L377: “p <”
L381: “fibre in diets had no effect on growth rate,”
Figure 1: In my opinion this figure could be deleted.
Tables: Please check all the tables. The error term should be reported with the same number of digits of the mean. Please uniform the number of digits also for p-values.
Table 3: In material and methods are reported 10 rabbits per group. Why in Table 3 there are 8 to 10 rabbit per group?
Table 7: In material and methods are reported 10 rabbits per group. Then the authors specify that seven samples were discarded, however, form the numbers reported in Table 7 it seems that 8 samples were discarded.
Author Response
We appreciate the comments made, which enriched the manuscript. The answer and comments to the suggestions made are in the file attached.
